# Adaptation and validation of the quality of contraceptive counseling (QCC) scale for use in Ethiopia and India

Kelsey Holt[1]*, Ewenat Gebrehanna[2], Shashi Sarnaik[1], Lakhwani Kanchan[3], Reiley Reed[1¤], Aman Yesuf[2], Bella Vasant Uttekar[3]

1 Department of Family and Community Medicine, University of California, San Francisco, San Francisco, California, United States of America, 2 School of Public Health, St. Paul's Hospital Millennium Medical College, Addis Ababa, Ethiopia, 3 Centre for Operations Research and Training, Vadodara, India

¤ Current address: School of Social Welfare, University of California, Berkeley, California, United States of America

* Kelsey.Holt@ucsf.edu

**Data Availability Statement:** The data that support the findings of this study are openly available at Dryad [https://datadryad.org/stash/share/wJe1DR

## Abstract

We adapted the Quality of Contraceptive Counseling (QCC) scale, originally constructed in Mexico, for Ethiopia and India to expand its utility for measurement of client experiences with counseling. Scale items were modified based on prior research on women's preferences for counseling in each country, and refined through cognitive interviews (n = 20 per country). We tested the items through client exit surveys in Addis Ababa, Ethiopia (n = 599), and Vadodara, India (n = 313). Psychometric analyses revealed the adapted scales were valid and reliable for use, and the final scales retained content validity according to the original published QCC construct definition. Specifically, confirmatory factor analysis revealed high factor loadings for almost all items on the original dimensions: Information Exchange, Interpersonal Relationship, Disrespect and Abuse. Internal consistency reliability was high in both settings (Alpha = 0.92 for QCC-Ethiopia and 0.74 for QCC-India). Final item pools contained 26 items in the QCC-Ethiopia Scale and 23 in the QCC-India Scale. Correlation analyses established convergent validity. QCC Scales and subscales fill a gap in measurement tools for ensuring high quality of care and fulfillment of human rights in contraceptive services, and consistent findings across continents suggest versatility in use across different contexts.

## Introduction

High quality, person-centered contraceptive counseling helps promote individuals' reproductive autonomy and well-being and is critical to ensuring the human rights principles of informed choice, non-discrimination, and autonomy are fulfilled in contraceptive services [1]. Recent research across various country settings has highlighted several threats to fulfillment of women's human rights in contraceptive care in the FP2020 era, including overly directive or coercive counseling and lack of information and counseling on side effects to support informed choices about contraceptive use [2–8].

wZ9JYQlLVL1yQq8aL7WpD8n0ba9aSMCcAAtlo; doi:10.7272/Q6FB516J)].

**Funding:** KH acknowledges funding support from the David & Lucile Packard Foundation (#2018-67680; www.packard.org). The funders had no role in study design, data collection and analysis, decision to publish, or preparation of the manuscript.

**Competing interests:** The authors have declared that no competing interests exist.

This emerging evidence base on deficiencies in counseling experienced by women in many settings suggests a need for more concerted counseling monitoring efforts in contraceptive programs. Over the last several years, there has been growing recognition of the limitations of client exit interview tools that have been deployed in programs since the early 1990's. These widely used tools focus primarily on information receipt rather than the broader counseling experience, do not probe directly for negative experiences, and have not been systematically evaluated for their validity and reliability [9, 10].

To help fill this gap in available measures, members of our team developed and validated a new client experience measure, the Quality of Contraceptive Counseling (QCC) Scale, grounded in human rights principles [11]. The QCC Scale is meant to provide a valid, reliable measure of client experiences with counseling along three dimensions: information exchange, interpersonal relationship, and presence of disrespect and abuse. It is inclusive of items that probe not only about information receipt but also degree of personalization and respectful treatment during counseling. The measure is applicable to all scenarios in which women interact with providers about contraception (including dedicated family planning visits for new or returning users, prenatal visits, post-abortion counseling, etc.), and produces comparable composite scores regardless of the depth of counseling provided or whether they chose to use a method.

The QCC-Mexico Scale was originally developed and validated in two states in Mexico. As a first step toward adapting the scale for use in other settings, our team conducted focus group studies with contraceptive clients in Ethiopia and India to understand women's expressed preferences for counseling in these settings [12, 13]. In this manuscript, we report on the adaptation of QCC scale items and findings from a survey study conducted for the purpose of testing the validity and reliability of the revised QCC Scales (QCC-India and QCC-Ethiopia) for use in quality and rights monitoring in both countries.

## Methods

### Overview

The original QCC-Mexico Scale items were adapted for the context of Ethiopia and India, with items translated into Amharic and Gujarati and several new items added for each country. Item pools were then fielded among contraception clients and psychometrically evaluated separately in each country to examine the extent to which the QCC-Ethiopia and QCC-India Scales replicated the factor structure and scale/item properties of the original QCC-Mexico scale. Full details on methodological approach to developing the original QCC-Mexico Scale are available elsewhere [11].

### Ethics statement

The University of California, San Francisco (UCSF) Institutional Review Board (IRB) reviewed and approved plans for this study in Ethiopia and deemed the study in India exempt due to the UCSF team's lack of access to participants' identifying information. In Ethiopia, the St. Paul's Hospital Millennium Medical College IRB reviewed and approved the study. In India, the Centre for Operations Research and Training (CORT) IRB reviewed and approved the study. In all study sites, each clinic's administrator granted permission for the study team to collect data.

Oral consent was obtained from all participants. All IRBs approved use of oral consent because written consent would constitute the only form of identifying information collected from study participants.

## Setting

In Ethiopia, our research took place in the capital city, Addis Ababa, in public health centers and franchise sites of a large non-governmental organization providing reproductive health services. In India, our research took place in Vadodara, the third largest city in the state of Gujarat, in government public health centers.

## Item adaptation: Contextualization and cognitive interviews

Each country's study team made a small set of initial modifications to the originally validated QCC items from Mexico to reflect the local context and women's preferences for contraceptive counseling identified in earlier formative work conducted for the purpose of QCC scale adaptation [12, 13], and translated the items into Amharic (Ethiopia) and Gujarati (India). We then conducted cognitive interviews with 20 women in each country to ensure that the scale items were relevant and comprehensible (recruitment approach and eligibility criteria for cognitive interviews was the same as that detailed below for the client exit interview surveys fielded to quantitatively validate the scale). Interviews consisted of administering the scale to participants and pausing after each question to ask participants to describe how and why they arrived at their answer, and whether the item was confusing or difficult to respond to. Cognitive interviews lasted on average 30 minutes in Ethiopia and 45 minutes in India.

Interviewers took hand-written notes for each item on participants' reasoning behind the answer they selected, inconsistencies in their response to that item compared to similar items, and participants' reflections on how easy or difficult it was to answer the item. These notes were later entered into Excel. Each country's study team then met to review results and make any necessary changes to items to ensure consistent interpretation and clarity of items. Study team members with fluency in each language then finalized item pools for testing.

## Scale validation: Client exit surveys

We conducted surveys with a convenience sample of contraception clients in both countries (N = 599 in Ethiopia and N = 313 in India) to test the final item pools. We aimed to have at least 10 participants per item, per general guidance for sample size calculations in confirmatory factor analysis (CFA) [14]; for our survey this translated to at least 300 given we anticipated no more than 30 items in each country's final scale. In Ethiopia, our sample size was larger than required to facilitate other planned analyses, beyond the scope of this scale validation study.

**Sample, recruitment, and data collection.** Eligibility criteria for the surveys included being female and having spoken with a provider about starting, changing, or discontinuing contraception either on the day of recruitment (in Ethiopia) or within the past two weeks (in India). Recognizing that client report of quality of counseling is likely clustered by provider, we recruited from a convenience sample of multiple health care settings in each country to encourage variability in client experience, as described below. In Addis Ababa, sites with known high volume of contraceptive clients were prioritized.

In Ethiopia, we recruited in four public health centers (n = 301 surveys) and four non-profit franchise sites (n = 298) in Addis Ababa, for a total sample size of N = 599 surveys. In the public health centers in each setting, family planning policy ensures availability of a full range of contraceptive methods (including oral contraceptive pills (OCPs), injectables, intrauterine devices (IUDs), implant, referral for sterilization, emergency contraception, and condoms) without cost, though stock outs can occur. In franchise sites, the full range of methods is available to clients for minimal cost. Survey recruitment took place in waiting rooms of study sites between May and June 2019. On recruitment days, data collectors approached all women who

appeared to be of reproductive age to invite them to participate in an exit interview about their experiences that day with contraceptive counseling. Clinic staff also gave information about the study to clients and directed them to study staff. After participants gave verbal consent, surveys were administered in private areas of clinics by study staff not affiliated with the clinic and took on average 30 minutes.

In India, we recruited in five health centers in urban areas (n = 154 surveys) and five in rural areas (n = 159 surveys) of Vadodara, for a total of N = 313 surveys. At the time of the study, public health centers in Gujarat provided sterilization, IUDs, OCPs, injectables, condoms and emergency contraceptive pills free of cost. Urban and rural locations were selected to capture experiences in different settings within the district. Survey recruitment took place in February and March 2020. The study team received lists of women who had sought contraceptive services in the prior two weeks from each health center. Women were selected purposively based on the type of method they were using in proportion to the commonality of each method in the list. Women were contacted at their homes to receive an invitation to participate in the study. After women provided verbal consent, surveys were administered in women's homes with efforts made by study staff to maintain privacy and confidentiality. Surveys took on average 35 minutes.

**Instrument.** The survey instrument in each setting included QCC Scale items, overall experience rating questions used to assess scale validity, and participant and visit background characteristics.

The QCC Scale was administered with a four-point response scale. As with the original scale, response categories for positively-worded items were "completely agree," "agree," "disagree," and "completely disagree." Response categories for negatively-worded items in Ethiopia retained the original categories "yes," "yes with doubts," "no with doubts," and "no;" while in India the wording was changed to "yes," "maybe yes," "maybe no," and "no," after conducting initial cognitive interviews and based on assessment of the local study team that this wording would be more appropriate for translation to Gujarati.

We assessed overall perception of the interaction with the provider by asking a general question about their experience with the provider, with response options on a four-point scale ranging from very good to very bad.

We asked women whether they would like to prevent a pregnancy (Yes, Unsure, No, Currently Pregnant), and whether they planned to use the method they selected that day, or continue the method they were already using (Yes, Unsure, No).

We collected information on women's age, education, occupation, number of children, insurance coverage, and marital status. We also asked participants about the type of provider they spoke to on the day of the survey (Ethiopia) or on their most recent visit in the last two weeks (India), the gender of the provider, and the reason for their visit.

**Analysis.** We used classical test theory (CTT) and CFA to test whether the original scale properties from Mexico held in Ethiopia and India and to examine properties of individual items comprising the separate versions of the QCC Scale in Ethiopia (QCC-Ethiopia) and India (QCC-India). CFA was chosen rather than exploratory factor analysis (EFA) due to the relatively small number of modifications made to the scale from the original version and our resulting hypothesis that the original dimensionality of the scale from Mexico would hold in Ethiopia and India. All analyses were conducted using Stata version 15.

These analyses were used in an iterative manner, and findings from each were triangulated to make final decisions about the item pool independently for each country. Considerations of content validity, drawing upon the QCC measurement framework [15] to make sure the full range of the construct was covered, were the most important factor when deciding whether an item would ultimately be removed from each scale after examining statistical analyses.

Convergent validity was then examined through analyses of the correlation of QCC Scale scores with other variables collected concurrently. We ran complete case analyses for all models.

Descriptive statistics, including category frequencies, means, standard deviations, were calculated for each item and subscale and for the composite scale scores for each version of the QCC Scale (QCC-Ethiopia, QCC-India). As with the original QCC Scale from Mexico, we calculated composite scores using a simple mean of all relevant item responses on the 4-point response scale and examined skew for all subscale and scale scores [11]. Internal consistency reliability was calculated using Cronbach's alpha. Excluded item alphas were assessed to see if removal of a given item changed the Cronbach's alpha score notably (more than 0.2). These analyses were all conducted both on subscale and overall QCC scores.

We conducted structured equation modelling via CFA to evaluate whether the original factor structure identified in Mexico was compatible with data from Ethiopia and India, and to identify potential items to remove for scale application in these new settings. Items not loading at least 0.4 on their assigned factor were considered for removal unless there was a strong justification from a content validity perspective to retain them [16]. We examined goodness of fit of CFA models for each country and calculated misfit indices to identify potential modifications to the covariance structure that would improve model fit. We also used Pearson correlation coefficients to examine the relationship between subscales.

We assessed convergent validity (i.e., the degree to which QCC Scale scores correlate with other similar measures as expected) using the overall measure of patient experience with the provider and the measure of whether they planned to use/continue using the selected contraceptive method (among those who reported selecting or already having a method on the day of the survey). As with the original validation study in Mexico, these analyses were conducted as bivariate logistic regressions with the dependent variables (experience and plan to use) dichotomized to highest score versus all else and the independent variable as a continuous QCC scale score. We conducted a sensitivity analysis treating the dependent experience variable as categorical to see whether this improved precision of the model after initial results produced wide confidence intervals.

## Inclusivity in global research

Additional information regarding the ethical, cultural, and scientific considerations specific to inclusivity in global research is included in S1 Checklist.

# Results

## Item adaptation

A table available on the University of California, San Francisco website, linked here, presents the differences in item pools between Mexico, Ethiopia, and India following formative research and cognitive interviewing in Ethiopia and India, and specifies reasons for modifications made. We link to https://qccscale.ucsf.edu/qcc-scale-versions rather than printing scale items in this manuscript due to copyright restrictions on the original publication of the QCC-Mexico scale (1). All original items from Mexico were included in the initial item pools for Ethiopia and India as they were deemed by study teams to be potentially relevant for the new contexts based on formative focus groups conducted prior to this study. As detailed in the table, four items were added to Ethiopia and India item pools to represent additional important areas of quality of counseling based on formative research, and one additional Disrespect and Abuse item was modified in India before cognitive testing to reflect the formative finding that providers scolding women due to their choice of methods was a more salient dimension of

discrimination than age-based discrimination. Two additional modifications were made to wording of items in India after cognitive testing and no changes were made in Ethiopia. In both countries, cognitive interviewing helped improve translations of original items into Amharic and Gujarati.

## Scale validation

**Participant and visit characteristics.** In both countries, we recruited a diverse sample in terms of both demographic characteristics (Table 1) and the context in which they received contraceptive counseling and their current use of a contraceptive method (Table 2).

**Psychometric analysis.** Table 3 shows the mean response on a scale of 1–4 (with 4 being the highest quality) and standard deviation for each scale item retained in QCC-Ethiopia and QCC-India scales and for the final subscale and overall scores for each version. Overall, scores were higher in India than in Ethiopia. As with the original scale in Mexico, QCC-Ethiopia and QCC-India scores tended towards higher quality, though variability in responses was observed for the majority of items. Skew of QCC-Ethiopia and QCC-India total scores was acceptable (–0.4 and –1.3 respectively) indicating the appropriateness of mean scores and retaining the 4-point response scale. Skew of Information Exchange and Interpersonal Relationship subscale scores was also acceptable (-0.1 and -0.2 for QCC-Ethiopia and -0.6 and -0.3 for QCC-India, respectively). As observed with the QCC-Mexico, the Disrespect and Abuse subscale was skewed (–6.6 for QCC-Ethiopia and -6.0 for QCC-India), suggesting that as a standalone measure of disrespect and abuse in family planning care, responses should be dichotomized to report of anything less than the highest score versus all other responses. The final QCC scale for use in India and Ethiopia, including item wording in English, Gujarati, and Amharic, can be accessed on the QCC website, linked to here (https://qccscale.ucsf.edu/qcc-scale-versions).

Cronbach's alpha was 0.74 for the full 23-item QCC-India Scale and 0.92 for the full 26-item QCC-Ethiopia Scale, suggesting acceptable internal consistency reliability in India and excellent reliability in Ethiopia (Table 3). Table 3 also displays Cronbach's alphas for the three subscales in each country, which were all acceptable. Excluded item alphas revealed that, in both countries, removal of no items changed subscale alphas more than 0.2, suggesting that the items measure the same underlying constructs as the other items in their respective subscales; additionally, no item's removal resulted in a change in more than 0.2 in the overall (full scale) alpha.

CFA revealed that both countries' data fit well with the three-factor structure identified in the original QCC scale construction in Mexico. We moved one item (prov_insist) from the Disrespect and Abuse subscale to the conceptually-related Interpersonal Relationship subscale after initial CFA modeling revealed very poor performance: subscale alphas were 0.4 and 0.6 and the item loaded 0.02 and 0.14 in Ethiopia and India, respectively, when the item was included on the Disrespect and Abuse subscale. Of the 23 items included in the final India model, 21 items loaded at least 0.4 onto their assigned factors; of the 26 items in Ethiopia, 25 met the same threshold (Table 3). Goodness of fit indices revealed moderate fit for the final CFA models: though likelihood ratios achieved statistical significance, indicating room for improvement, RMSEA, comparative fit index, and Tucker-Lewis index values were favorable [17].

Examination of the Pearson correlations coefficients between the subscale scores showed that the relationships observed in Mexico largely remained the same in Ethiopia, providing evidence for a single unifying latent QCC construct in this setting; specifically, the Information Exchange subscale and Interpersonal Relationship subscales were highly correlated (0.8 in Ethiopia, similar to the 0.7 observed in Mexico) and the Disrespect and Abuse subscale

**Table 1. Participant demographic characteristics.**

| Characteristic | |
|---|---|
| | **Ethiopia[1] (n = 599)** |
| Age, mean (SD, range) | 27 (5.5, 15–47) |
| Marital Status, n (%) | |
| Never married | 118 (19.7) |
| Married or living together | 464 (77.6) |
| Divorced, separated, or widowed | 16 (2.6) |
| Religion, n (%) | |
| Orthodox Christian | 470 (78.5) |
| Muslim | 62 (10.4) |
| Protestant | 67 (11.2) |
| Number of living children, median (IQR, range) | 2 (2, 0–12) |
| Education, n (%) | |
| No schooling | 73 (12.2) |
| Primary | 204 (34.1) |
| Secondary | 209 (34.9) |
| Licensing or professional school (TEVT) | 57 (9.5) |
| College or above | 56 (9.3) |
| Occupation, n (%)[2] | |
| Worked for payment | 225 (37.6) |
| Worked for self | 64 (10.7) |
| Helped with a family business, with no payment | 11 (1.8) |
| Household chores, childcare | 344 (57.4) |
| No work | 36 (6.0) |
| Study | 28 (4.7) |
| | **India[3] (n = 313)** |
| Age, mean (SD, range) | 28 (4.4, 19–47) |
| Region, n (%) | |
| Rural | 159 (50.8) |
| Urban | 154 (49.2) |
| Marital Status, n (%) | |
| Never married | 4 (1.3) |
| Married | 309 (98.7) |
| Religion, n (%) | |
| Hindu | 257 (82.1) |
| Muslim | 56 (17.9) |
| Number of living children, Median (IQR, range) | 2 (1, 1–5) |
| Education (years of schooling completed), n (%) | |
| No schooling | 18 (5.8) |
| Fewer than 5 years | 17 (5.4) |
| 5–7 years | 62 (19.8) |
| 8–9 years | 65 (20.8) |
| 10–11 years | 74 (23.6) |
| 12+ years | 77 (24.6) |
| Occupation, n (%)[1] | |
| Worked for payment | 25 (8.0) |
| Worked for self | 28 (8.9) |

*(Continued)*

**Table 1.** (Continued)

| | |
|---|---|
| Helped with a family business, with no payment | 2 (0.6) |
| Household chores, childcare | 299 (95.5) |

[1] Missing data ranged from 0–2 depending on the variable

[2] Participants could select more than one response option

[3] No missing data

correlated 0.3 with both Information Exchange and Interpersonal Relationship in Ethiopia compared to the 0.3 and 0.4, respectively, observed in Mexico [11]. In India, the Information Exchange and Interpersonal Relationship subscales were correlated similarly to the other two countries (0.7); however, the Disrespect and Abuse subscale was less correlated with the other subscales (0.1 with Information Exchange and 0.05 with Interpersonal Relationship).

## Correlational validity

In both countries, full scale scores were highly correlated with a dichotomous overall measure of patient experience (Ethiopia: OR = 131.4, 95% CI = 52.0–332.7; India: OR = 1.8, 95% CI = 1.8–15.0) (Table 4). Subscale scores were also significantly correlated with the overall experience measure in both countries, with the exception of the Disrespect and Abuse subscale in India. In Ethiopia, among participants who had selected a method after their interaction with a provider about contraception, there was also correlation between QCC-Ethiopia Scale overall and subscale scores and their reported intention to initiate use of the method. We did not conduct this analysis for India because all participants had begun using a method by virtue of our recruitment approach (wherein we conducted interviews on a later date within two weeks after the consultation).

## Discussion

In this manuscript, we provide evidence for the validity and reliability of adapted versions of the QCC Scale for use in Ethiopia (QCC-Ethiopia; Amharic language, tested with data from Addis Ababa) and India (QCC-India; Gujarati language, tested with data from Vadodara). The final QCC-Ethiopia Scale is 26 items and the final QCC-India version has 23 items, compared to 22 items in the original QCC-Mexico Scale (Spanish language). The large majority of the original items from Mexico were retained in the two new countries after formative research and field testing revealed they were appropriate across contexts, and the new scale versions retained the original factor structure with three underlying dimensions: Information Exchange, Interpersonal Relationship, and Disrespect and Abuse. One of the original Disrespect and Abuse items (prov_insist) was moved to the conceptually-related Interpersonal Relationship subscale as this greatly improved the alphas for the Disrespect and Abuse subscale and the factor loading for the item in India (though not Ethiopia, where it was nonetheless retained due to the critical importance of capturing potential pressure from providers from a content validity perspective).

Minor differences between the three countries' QCC Scale versions represent contextual differences in what emerged as the most salient manifestations of each sub-domain of the construct in each country. In the Information Exchange subscale, the item "personal" was added to the QCC-Ethiopia and QCC-India scales to reflect that providers asking clients personal questions was a critical way identified in formative research for women to feel their counseling experience is personalized. The item "explain" was removed from the QCC-India Scale as it

**Table 2. Participant contraceptive choices and preferences and visit characteristics, n (%).**

| | Ethiopia (n = 599) | India (n = 313) |
|---|---|---|
| Would you like to avoid pregnancy right now?[1] | | |
| Yes | 532 (89.1) | 306 (97.8) |
| Unsure | 14 (2.3) | 2 (0.6) |
| No | 43 (7.2) | 4 (1.3) |
| N/A–I am pregnant | 8 (1.3) | 1 (0.3) |
| Were you using a contraceptive method when you came to your consultation? If yes, please indicate the method you were using. | | |
| Method using | 413 (68.9) | 110 (35.1) |
| *Pill* | *64 (15.5)* | *25 (22.7)* |
| *Injectable* | *214 (51.8)* | *8 (7.5)* |
| *IUD* | *29 (7.0)* | *9 (8.2)* |
| *Implant* | *94 (22.8)* | *0 (0.0))* |
| *Condoms* | *1 (0.2)* | *66 (60.0)* |
| *Female sterilization* | *0 (0.0)* | *2(1.8)* |
| *Emergency contraception* | *2 (0.5)* | *0(0.0)* |
| *Other (unspecified)* | *9 (2.2)* | *-* |
| Not currently using a method | 186 (31.1) | 203 (64.9) |
| Did you or your provider decide on a method to use during this consultation?[2] If so, which method? | | |
| Method chosen | 511 (86.8) | 299 (95.5) |
| *Pill* | *62 (12.1)* | *55 (17.6)* |
| *Injectable* | *270 (52.8)* | *6 (1.9)* |
| *IUD* | *21 (4.1)* | *63 (20.1)* |
| *Implant* | *105 (20.5)* | *0 (0)* |
| *Condoms* | *1 (0.2)* | *88 (28.1)* |
| *Female sterilization* | *1 (0.2)* | *79 (25.2)* |
| *Emergency contraception* | *1 (0.2)* | *0 (0.0)* |
| *Other (unspecified)* | *50 (9.8)* | *8 (2.6)* |
| No, I did not decide on a method | 78 (13.2) | 14 (4.5) |
| Would you like to use (or plan to continue using) the method you selected or had placed during this consultation?[3] | | |
| Yes | 314 (54.0) | 186 (59.4) |
| Unsure | 134 (23.0) | 94 (30.0) |
| No | 29 (5.0) | 24 (7.7) |
| N/A, didn't receive a method | 105 (18.0) | 9 (2.9) |
| What type of provider did you speak with?[4] | | |
| Doctor | 20 (3.4) | 104 (33.2) |
| Nurse | 101 (17.0) | 55 (17.6) |
| Health Officer | 4 (0.7) | - |
| Lay health worker (LHV) | - | 10 (3.2) |
| Auxiliary nurse midwife (ANM) | - | 50 (16.0) |
| Accredited social health activist (ASHA) or Urban social health activist (USHA)[5] | - | 299 (95.5) |
| Unsure | 469 (79.0) | - |
| Other (unspecified) | - | 2 (0.6) |
| What was the sex of the provider you saw?[6] | | |
| Female | 586 (98.7) | 279 (89.1) |

(*Continued*)

**Table 2.** (Continued)

|  | Ethiopia (n = 599) | India (n = 313) |
|---|---|---|
| Male | 8 (1.4) | 34 (10.9) |
| What was the reason for your visit?[7] |  |  |
| Request a contraceptive method | 389 (64.9) | 37 (11.8) |
| Ask for information | 120 (20.0) | 249 (79.6) |
| Removal of method | 95 (15.9) | 1 (0.3) |
| Method follow-up | 170 (28.4) | 94 (30.0) |
| Abortion or post-abortion care | 37 (6.2) | 0 (0) |
| Other[8] | 32 (3.7) | 1 (0.3) |

[1]This variable was missing 2 responses in Ethiopia.

[2] This variable was missing 10 responses in Ethiopia.

[3] This variable was missing 17 responses in Ethiopia.

[4] Respondents could select more than one response option in India. In Ethiopia, this variable was constrained to one answer option and was missing 5 responses.

[5]ASHAs and USHAs are government-trained and funded community health workers. The ASHAs provide care in rural regions, while USHAs provide care in urban areas.

[6] This variable was missing 5 responses in Ethiopia.

[7] Respondents could select more than one response option.

[8] Reasons in Ethiopia included ART follow-up, primary care, HIV testing, pregnancy test, treatment for bleeding, or vaccination (two were unspecified). In India, the one "Other" was unspecified.

did not load well with other items as it had in Mexico and Ethiopia, suggesting women's perception of their provider's ability to explain contraception is not central to determining the degree of quality they perceive. "Explain" was not deemed critical to the Information Exchange sub-domain given that many other items cover whether participants perceived receiving sufficient and understandable information about contraception.

In the Interpersonal Relationship subscale, the items "express_self" (reflecting women's opportunities to express opinions and concerns) and "no_interrupt" (reflecting providers' attempts to ensure no interruptions) were added and retained in the QCC-Ethiopia and QCC-India versions as additional important manifestations of what it means for a provider to develop a positive, trusting relationship with clients. In India, "prov_friendly," and "prov_knows" were removed as less critical manifestations of the Interpersonal Relationship domain; friendliness is not integral to our definition of high-quality counseling [15] and may be a context specific expectation for high quality counseling not relevant in India. "Prov_knows" was an Interpersonal Relationship item developed out of formative research in Mexico that suggested clients have more trust in providers they deem knowledgeable; while this item appeared to work well in Ethiopia, its lack of correlation with other sub-domain items in India suggests this may be another context-specific interpretation of what it means to have a trusting relationship with a provider. For the final scale domain, Disrespect and Abuse, a new item was added and retained ("scold_marital") to reflect the salience of discrimination in counseling based on marital status in both QCC-Ethiopia and QCC-India Scales. Additionally, the item "scold_age" was modified to "scold_use" for India to reflect the greater salience of discrimination based on method choice compared to one's age in this setting.

The two items in India and one item in Ethiopia retained in final scales with loadings below the 0.4 threshold reflect important concepts from a content validity standpoint. As mentioned, the Disrespect and Abuse item "prov_insist" was retained in Ethiopia despite a low factor

**Table 3. Psychometric analysis results for the Quality of Contraceptive Counseling (QCC) scale from Ethiopia (QCC-Ethiopia; N = 599) and India (QCC-India; N = 313)[1].**

| | QCC-Ethiopia | | | | QCC-India | | | |
|---|---|---|---|---|---|---|---|---|
| | CFA[2] coeff-icient | Mean (SD[2]) | Excluded-item alphas | | CFA[2] coeff-icient | Mean (SD[2]) | Excluded-item alphas | |
| | | | Full scale | Subscale | | | Full scale | Subscale |
| **Information exchange[3]** | | 2.9 (0.5) SK$^2$ = -0.1 | | | | 3.3 (0.3) SK$^2$ = -0.6 | | |
| opinion | 0.51 | 3.3 (0.6) | 0.91 | 0.89 | 0.72 | 3.4 (0.5) | 0.71 | 0.77 |
| personal | 0.63 | 3.1 (0.6) | 0.91 | 0.88 | 0.63 | 3.3 (0.6) | 0.71 | 0.77 |
| info | 0.51 | 2.4 (0.8) | 0.91 | 0.88 | 0.42 | 3.4 (0.5) | 0.73 | 0.80 |
| explain | 0.64 | 3.2 (0.6) | 0.91 | 0.88 | - | - | - | - |
| opportunity | 0.56 | 3.2 (0.6) | 0.91 | 0.88 | 0.18 | 3.4 (0.5) | 0.74 | 0.82 |
| sti_info | 0.49 | 2.5 (0.8) | 0.91 | 0.88 | 0.62 | 3.1 (0.6) | 0.72 | 0.78 |
| method_fail | 0.73 | 2.8 (0.7) | 0.91 | 0.87 | 0.60 | 3.2 (0.7) | 0.72 | 0.78 |
| body_react | 0.76 | 2.9 (0.7) | 0.91 | 0.87 | 0.60 | 3.2 (0.6) | 0.72 | 0.78 |
| method_use | 0.74 | 3.0 (0.6) | 0.91 | 0.87 | 0.49 | 3.3 (0.5) | 0.73 | 0.79 |
| method_react | 0.74 | 2.8 (0.7) | 0.91 | 0.87 | 0.65 | 3.1 (0.7) | 0.71 | 0.77 |
| method_stop | 0.74 | 2.9 (0.7) | 0.91 | 0.87 | 0.47 | 3.3 (0.5) | 0.73 | 0.79 |
| **Subscale alpha** | | | | 0.89 | | | | 0.80 |
| **Interpersonal relationship** | | 3.1 (0.4) SK$^2$ = -0.2 | | | | 3.2 (0.3) SK$^2$ = -0.3 | | |
| info_private | 0.56 | 3.1 (0.6) | 0.91 | 0.82 | 0.12 | 3.6 (0.5) | 0.74 | 0.74 |
| enough_time | 0.68 | 3.0 (0.6) | 0.91 | 0.81 | 0.58 | 3.3 (0.5) | 0.72 | 0.67 |
| prov_friendly | 0.65 | 3.4 (0.6) | 0.91 | 0.81 | - | - | - | - |
| prov_knows | 0.70 | 3.2 (0.5) | 0.91 | 0.81 | - | - | - | - |
| prov_health | 0.66 | 2.9 (0.7) | 0.91 | 0.81 | 0.57 | 3.3 (0.5) | 0.72 | 0.68 |
| prov_opinions | 0.73 | 3.0 (0.6) | 0.91 | 0.80 | 0.63 | 3.3 (0.5) | 0.72 | 0.67 |
| express_self | 0.75 | 2.9 (0.7) | 0.91 | 0.80 | 0.69 | 3.0 (0.7) | 0.71 | 0.65 |
| prov_listens | 0.70 | 3.1 (0.6) | 0.91 | 0.80 | 0.59 | 3.3 (0.5) | 0.72 | 0.68 |
| no_interrupt | 0.51 | 3.0 (0.7) | 0.91 | 0.82 | 0.65 | 3.2 (0.7) | 0.72 | 0.65 |
| prov_insist[4] | 0.11 | 3.6 (0.4) | 0.92 | 0.87 | 0.49 | 3.0(1.3) | 0.83 | 0.73 |
| **Subscale alpha** | | | | 0.83 | | | | 0.71 |
| **Disrespect and abuse** | | 4.0 (0.2) SK$^2$ = -6.6 | | | | 3.9 (0.4) SK$^2$ = -6.0 | | |
| prov_judge | 0.46 | 3.9 (0.4) | 0.92 | 0.70 | 0.43 | 3.8 (0.6) | 0.75 | 0.85 |
| scold_age | 0.60 | 3.9 (0.3) | 0.92 | 0.64 | - | - | - | - |
| scold_use | - | - | - | - | 0.54 | 3.9 (0.6) | 0.73 | 0.81 |
| prov_sexlife | 0.61 | 4.0 (0.3) | 0.92 | 0.63 | 0.96 | 3.9 (0.4) | 0.74 | 0.73 |
| prov_touched | 0.58 | 4.0 (0.2) | 0.92 | 0.68 | 0.86 | 3.9 (0.4) | 0.74 | 0.74 |
| scold_marital | 0.72 | 4.0 (0.2) | 0.92 | 0.64 | 0.93 | 3.9 (0.4) | 0.74 | 0.73 |
| **Subscale alpha** | | | | 0.71 | | | | 0.81 |
| **Full QCC scale** Mean(SD) | | 3.2 (0.4) | | | | 3.4 (0.2) | | |
| SK | | -0.4 | | | | -1.3 | | |
| Alpha | | 0.92 | | | | 0.74 | | |
| **CFA goodness of fit statistics[5]** | | | | | | | | |
| Likelihood ratio | 1310 (p<0.001) | | | | 444 (p<0.0001) | | | |
| RMSEA | 0.076 (p<0.001) | | | | 0.055 (p = 0.12) | | | |
| Comparative fit index | 0.86 | | | | 0.92 | | | |

*(Continued)*

**Table 3.** (Continued)

|  | **QCC-Ethiopia** |  | **QCC-India** |  |
|---|---|---|---|---|
| Tucker-Lewis index | 0.85 |  | 0.91 |  |

[1] Missing data varied by item, ranging from 0 to 10 per item in Ethiopia and 0 in India. Factor analysis was run using the Stata "MLMV" option which uses all available observations, accounting for missing at random (MAR) data.

[2] CFA = Confirmatory factor analysis; SD = standard deviation; SK = skew

[3] Please refer to the QCC website, https://qccscale.ucsf.edu/qcc-scale-versions, for item wording

[4] This item loaded as part of the Disrespect and Abuse subscale in the original scale development in Mexico [11]. In our CFA modeling it performed very poorly on that subscale (subscale alphas were 0.4 and 0.6 and the item loaded 0.02 and 0.14 in Ethiopia and India, respectively) and thus was moved to the conceptually-related Interpersonal Relationship subscale, as shown here.

[5] After examining modification indices from the CFA model in Ethiopia, we adjusted the model to allow the error terms for the STI_info and info variables to correlate with each other. (The modification index was 530 compared to the rest which were under 100). No major changes were suggested by modification indices in India so we did not make any modifications to the original model. SRMR was not able to be calculated due to the estimation technique which accounted for missing data.

loading because lack of pressure to use contraception is a critical component of the QCC conceptual framework. Low loading likely reflects the fact that pressure is a unique element of individuals' experiences not as correlated with other interpersonal aspects of counseling as observed in the original Mexico sample or in our India sample. In India, the Information Exchange item "opportunity" was retained because clients having the opportunity to be an active participant in method selection is a critical component of the construct not otherwise covered by other items, and the item "info_private" is also critical as privacy is a core component of the QCC construct not otherwise covered [15]. The low loadings of these items in India may reflect that, in this setting, these elements are not as central to their respective subdomains as observed in Mexico and Ethiopia samples.

Taken together, our psychometric analysis of QCC scale and item properties in Addis Ababa, Ethiopia and Vadodara, India suggest both the QCC-Ethiopia and QCC-India Scales are sufficiently valid and reliable for use in these new settings. Our strongest findings come from the CFA analyses which clearly demonstrate consistent dimensionality of the scale across contexts with excellent factor loadings for all but a few items retained for content validity purposes. Internal consistency reliability was also high in both countries for both overall and

**Table 4. Convergent validity results from logistic regression predicting the odds of related variables based on continuous[1] QCC-India and QCC-Ethiopia scale scores.**

|  | n (%) | Information Exchange subscale | Interpersonal Relationship subscale | Disrespect and Abuse subscale[1] | Full scale |
|---|---|---|---|---|---|
|  |  | OR (95% CI) | OR (95% CI) | OR (95% CI) | OR (95% CI) |
|  |  | p-value | p-value | p-value | p-value |
| Highest rating of overall experience with provider: India (n = 313) | 152 (49%) | 3.0 (1.5, 5.9) | 3.7 (1.5, 8.9) | 1.5 (0.8, 3.0) | 1.78 (1.8, 15.0) |
|  |  | P = 0.002 | P = 0.004 | P = 0.234 | P = 0.002 |
| Highest rating of overall experience with provider: Ethiopia (n = 598) | 163 (27%) | 17.2 (9.6, 30.8) | 49.8 (24.0, 103.3) | 3.2 (1.1, 9.1) | 131.4 (52.0, 332.7) |
|  |  | p<0.0001 | p<0.0001 | P = 0.033 | p<0.0001 |
| Intention to use method selected at baseline: Ethiopia[2] (n = 477) | 314 (66%) | 2.0 (1.3, 3.2) | 3.1 (1.8, 5.1) | 2.2 (1.0, 4.6) | 3.4 (1.9, 6.3) |
|  |  | p = 0.001 | p<0.0001 | p = 0.045 | p<0.0001 |

[1] Disrespect and Abuse score dichotomized into highest score (higher = better quality) versus all else, due to high skew.

[2] We did not examine this outcome in India because participants were interviewed within two weeks of their visit, with those initiating method use prioritized for recruitment. Missing data for this outcome are due to some women not having selected a method during their visit in Ethiopia.

subscale scores [18]. We also found good evidence of convergent validity, suggesting the QCC-Ethiopia and QCC-India Scales are measuring the intended construct.

Three versions of the scale, constructed with data from Mexico City and San Luis Potosi, Mexico (QCC-Mexico; Spanish) [11], Addis Ababa, Ethiopia (QCC-Ethiopia; Amharic), and Vadodara, India (QCC-India; Gujarati), are now tested and available for robust quality and human rights monitoring in contraception programs and research into the determinants and outcomes associated with high quality counseling. We encourage researchers and program managers to select the version that most closely matches their setting, and undergo additional translations and/or adaptations of the scale as necessary to best meet their needs. The fact that our adaptation and validation processes resulted in three versions of the QCC Scale that are highly consistent between settings suggests that, while the tailored versions of the scale that we present in this manuscript offer the benefit of having a menu of QCC Scale options to choose from, the QCC Scale is quite robust to contextual differences and may be easily transferable to other settings without extensive modifications. Though the QCC Scale was designed primarily to be a tool for facilities and systems to internally monitor quality through exit interviews with clients, it also has potential as a tool for social accountability or mystery client studies. A short form of the QCC Scale—the 10-item QCC-10—has also been developed and validated, which may be useful in population level studies of women's experiences with quality of counseling [19, 20].

The QCC Scale offers the benefit of producing both composite scores, allowing for a comprehensive look at women's experiences with contraception providers and examination of trends over time or between groups (e.g., by age or between returning versus new users), as well as subscale and individual item scores that allow for homing in on specific areas for attention. Further, the Disrespect and Abuse subscale is unique among other counseling quality measurement tools and can provide an accountability mechanism to ensure voluntarism and lack of abuse in settings where there is buy-in for monitoring for negative experiences [10]. Due to the skewed nature of Disrespect and Abuse scores in both countries (as well as the original validation study in Mexico (1)), we recommend they be dichotomized when used as stand-alone measures. It is also worth noting that the Disrespect and Abuse subscale was less correlated with other subscales in our India sample compared to the Ethiopia sample and what had previously been observed in the QCC-Mexico validation study. This suggests that in the Vadodara context, the experience of Disrespect and Abuse is more weakly tied to the overall QCC construct; in other words, a person's experience of extreme negative provider behaviors is less correlated with how they perceived counseling overall compared to other settings. Nonetheless, composite scores can still be computed and applied to capture the full range of the QCC construct.

Our findings are limited by our sampling only one geographic area in each country. It is unclear to what extent the QCC Scale is valid for use outside of Addis Ababa, Ethiopia, and Gujarat, India, and we recommend future studies examine the properties of the QCC scale in new settings. However, our finding that the three-dimensional QCC construct identified in Mexico was replicated in two different countries from two new continents with relatively consistent item pools suggests the scale is likely robust to other contexts as well. We also note that because providers in the study sites were aware the study was taking place, it is possible that they adjusted their behavior to provide better counseling; this could have inflated QCC scores.

## Conclusion

Adapted versions of the original QCC-Mexico Scale were found to be valid and reliable for use in measuring women's experiences with contraceptive counseling in Addis Ababa, Ethiopia

(the QCC-Ethiopia Scale), and Vadodara, India (the QCC-India Scale). Three related dimensions of the QCC construct—Information Exchange, Interpersonal Relationship, and Disrespect and Abuse—were replicated across contexts and comprehensively cover women's experiences of quality and rights in contraceptive counseling, with minor modifications to items to tailored for each setting. The QCC Scales provide a valuable tool for researchers and program managers to measure overall experience with counseling as well as individual domains or specific areas for counseling improvement. The consistency of items between the three country settings resulting from our extensive scale development and validation process across three continents suggests that, while context-specific modifications bolstered the validity of the scale for each site, the QCC construct is relatively robust to different settings.

## Supporting information

**S1 Checklist. Global health questionnaire.**
(DOCX)

## Acknowledgments

We would like to acknowledge the following individuals who helped with study coordination and data management: Jashoda Sharma, Fanna Adugna, Hailemichael Bizuneh, and Bekalu Assamnew. We would also like to acknowledge all the data collectors and field supervisors for their contributions. We would also like to thank Danielle Hessler Jones, Christine Dehlendorf, Erin Wingo, and Martha Michel for input into our data analysis process.

## Author Contributions

**Conceptualization:** Kelsey Holt, Ewenat Gebrehanna, Bella Vasant Uttekar.

**Formal analysis:** Kelsey Holt, Shashi Sarnaik, Aman Yesuf.

**Funding acquisition:** Kelsey Holt.

**Investigation:** Lakhwani Kanchan.

**Methodology:** Kelsey Holt, Shashi Sarnaik, Aman Yesuf.

**Supervision:** Ewenat Gebrehanna, Reiley Reed.

**Writing – original draft:** Kelsey Holt, Shashi Sarnaik.

**Writing – review & editing:** Kelsey Holt, Ewenat Gebrehanna, Shashi Sarnaik, Lakhwani Kanchan, Reiley Reed, Aman Yesuf, Bella Vasant Uttekar.

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
