## [Decision Letter · Decision Letter 0]

20 Jul 2022

PONE-D-21-36673Adaptation and validation of the quality of contraceptive counseling (QCC) scale for use in Ethiopia and IndiaPLOS ONE

Dear Dr. Holt,

Thank you for submitting your manuscript to PLOS ONE. After careful consideration, we feel that it has merit but does not fully meet PLOS ONE’s publication criteria as it currently stands. Therefore, we invite you to submit a revised version of the manuscript that addresses the points raised during the review process.

We look forward to receiving your revised manuscript.

Kind regards,

Nguyen Toan Tran

Academic Editor

PLOS ONE

Journal Requirements:

5. Please note that in order to use the direct billing option the corresponding author must be affiliated with the chosen institute. Please either amend your manuscript to change the affiliation or corresponding author, or email us at plosone@plos.org with a request to remove this option.

Reviewers' comments:

Reviewer's Responses to Questions

**Comments to the Author**

1. Is the manuscript technically sound, and do the data support the conclusions?

Reviewer #1: Yes

Reviewer #2: Yes

2. Has the statistical analysis been performed appropriately and rigorously? 

Reviewer #1: Yes

Reviewer #2: Yes

3. Have the authors made all data underlying the findings in their manuscript fully available?

Reviewer #1: Yes

Reviewer #2: Yes

4. Is the manuscript presented in an intelligible fashion and written in standard English?

Reviewer #1: Yes

Reviewer #2: Yes

5. Review Comments to the Author

Reviewer #1: The manuscript is an interesting piece of work. It is well written; however, I would like to forward the the following points to further refine the manuscript.

1.) If you have an interest of comparing an underlying factor structures in the two countries, the manuscript didn't serve its purpose as multigroup CFA is missing from the manuscript. If otherwise, what is the point of separately reporting findings of the two countries without making an objective assessment (statistical comparison)?

2.) In line 125 to 126, it is indicated that clinic staffs are engaged in the process of sample selection and are aware of the study objective and its process in the study from Ethiopia. This more likely would have an impact on the result of the study (introducing bias) as providers may get to be at their best behavior during the study period.

3.) In line 244-45, the authors mentioned that scores were skewed to higher quality and yet they used arithmetic mean as a summary measure which is highly influenced by skewed observation. Besides, the measurement is an ordinal scale and the use of arithmetic mean is not advisable in this scale of measurement.

4.) In line 252, the authors mentioned that they used MLVM command to run the factor analysis in STATA. MLVM is not a command; rather an option within the SEM command of STATA.

5.) In the reliability and validity analysis, the item "prov_insist" was kept as part of the third domain in Ethiopia and "opportunity" and "info_private" as part of the second domain. The reliability of these factors would have been improved if this items were removed. In addition, the items load poorly to their respective domains. This suggests that the authors should respecify the model and test a an alternative four or five factor CFA model (eg. pressure as a fourth domain) than retain then as part of the second or third.

6.) The result section has a component that should have been taken to the discussion section (Lines 278 to 303).

Reviewer #2: The authors present a pertinent and well-founded justification for carrying out the investigation.

With regard to the structure of the article, it has a careful presentation, is written in English, easy to read, with a logical sequence and interconnection between the contents. Contains all mandatory elements for publications in PLOS ONE.

The methodology used fits the type of study developed. It presents all the steps that it integrates in a very detailed, clear and robust way. The inclusion and exclusion criteria of the participants, the type of and its calculation are explained. The tests used for the adaptation and validation of the Quality of Contraceptive Counseling (QCC) Scale for use in Ethiopia and India are adjusted and appropriate. The authors justify the robust way and, based on the statistical analysis performed, the inclusion of items on the scale (even those with a low factor), as well as the exclusion of others.

They present results of an original research and with high methodological rigor.

The discussion is well founded, supported by the data obtained through the results.

Conclusions lead us to the importance and relevance of using a valid and reliable scale as an instrument to be used in quality contraceptive counseling.

Most references are from the last 5 years.

The study respected the ethical principles required for studies with human beings: the authorizations of the University of California and the respective countries (Ethiopia and Egypt), as well as the informed consent given by the study participants.

I consider the article relevant for publication, due to the importance of dissemination, use of the scale and the methodological robustness presented.

6. PLOS authors have the option to publish the peer review history of their article (what does this mean?). If published, this will include your full peer review and any attached files.

Reviewer #1: No

Reviewer #2: **Yes: **Sara Elisabete Cavaco Palma

---

## [Author Response · Author response to Decision Letter 0]

3 Oct 2022

Reviewer #1: The manuscript is an interesting piece of work. It is well written; however, I would like to forward the the following points to further refine the manuscript.

1.) If you have an interest of comparing an underlying factor structures in the two countries, the manuscript didn't serve its purpose as multigroup CFA is missing from the manuscript. If otherwise, what is the point of separately reporting findings of the two countries without making an objective assessment (statistical comparison)?

• Thank you for this clarifying question which helps us to refine the language we use in the manuscript to describe the QCC Scale versions. Our intention was not to compare the underlying factor structures between countries, but rather to assess independently the degree to which the two new scales replicated the factor structure and scale/item properties of the original QCC-Mexico scale. To make this more clear, we have made edits to the language throughout the entire manuscript to make clear that the QCC-Ethiopia and QCC-India scales were independently evaluated and that we consider these two separate, independently-assessed scales. 

2.) In line 125 to 126, it is indicated that clinic staffs are engaged in the process of sample selection and are aware of the study objective and its process in the study from Ethiopia. This more likely would have an impact on the result of the study (introducing bias) as providers may get to be at their best behavior during the study period.

• Thank you for this point. We have added in line 414 an additional sentence to the Limitations: “We also note that because providers in the study sites were aware the study was taking place, it is possible that they adjusted their behavior to provide better counseling; this could have inflated QCC scores.”

3.) In line 244-45, the authors mentioned that scores were skewed to higher quality and yet they used arithmetic mean as a summary measure which is highly influenced by skewed observation. Besides, the measurement is an ordinal scale and the use of arithmetic mean is not advisable in this scale of measurement.

• Thank you for this helpful point. We have added information on the skew of scale and subscale scores to the Results and Table 3. Because D&A subscale scores were highly skewed, we re-ran the correlation analyses to use a dichotomous (top-scored) measure, as we had previously done in the validation study in Mexico (and updated Table 4 accordingly). We clarified in the Discussion that because of the skew of the D&A scores we recommend they be dichotomized when used as a standalone measure.

• Re: to the original line pointed out by this reviewer, in line 256 we have changed the word “skewed” to “tended” to more precisely represent the point, since we did not find evidence of skew other than that related to the D&A subscale. 

• We have otherwise retained use of mean scoring per our understanding of the literature supporting use of mean scores for Likert scale data; see for example: 

o Sullivan GM, Artino AR Jr. Analyzing and interpreting data from likert-type scales. J Grad Med Educ. 2013 Dec;5(4):541-2. doi: 10.4300/JGME-5-4-18. PMID: 24454995; PMCID: PMC3886444.

o Norman G. Likert scales, levels of measurement and the "laws" of statistics. Adv Health Sci Educ Theory Pract. 2010 Dec;15(5):625-32. doi: 10.1007/s10459-010-9222-y. Epub 2010 Feb 10. PMID: 20146096.

4.) In line 252, the authors mentioned that they used MLVM command to run the factor analysis in STATA. MLVM is not a command; rather an option within the SEM command of STATA.

• Thank you for this point; we have changed the word “command” to “option” accordingly.

5.) In the reliability and validity analysis, the item "prov_insist" was kept as part of the third domain in Ethiopia and "opportunity" and "info_private" as part of the second domain. The reliability of these factors would have been improved if this items were removed. In addition, the items load poorly to their respective domains. This suggests that the authors should respecify the model and test a an alternative four or five factor CFA model (eg. pressure as a fourth domain) than retain then as part of the second or third.

• Thank you for this suggestion. Per this reviewer’s first point, we have modified language to make more clear that our goal with this analysis was to examine the extent to which the QCC-Ethiopia and QCC-India Scales replicated the factor structure and scale/item properties of the original QCC-Mexico scale (rather than using exploratory factor analysis to determine the underlying factor structure of each). Because the QCC-Ethiopia and QCC-India scales contained only minor item modifications compared to the original QCC-Mexico Scale, we feel this approach is justified and that the utility of these adapted versions of the QCC Scale is higher with a consistent factor structure between countries such that the three QCC subscales can be applied in different settings.

6.) The result section has a component that should have been taken to the discussion section (Lines 278 to 303).

• Thank you for this helpful suggestion. We have moved this text to the discussion section –integrated into paragraphs 2-4—accordingly (with the exception of the sentence reporting findings from the model fit exercise which we deemed appropriate to retain in Results).

Reviewer #2: The authors present a pertinent and well-founded justification for carrying out the investigation.

With regard to the structure of the article, it has a careful presentation, is written in English, easy to read, with a logical sequence and interconnection between the contents. Contains all mandatory elements for publications in PLOS ONE.

The methodology used fits the type of study developed. It presents all the steps that it integrates in a very detailed, clear and robust way. The inclusion and exclusion criteria of the participants, the type of and its calculation are explained. The tests used for the adaptation and validation of the Quality of Contraceptive Counseling (QCC) Scale for use in Ethiopia and India are adjusted and appropriate. The authors justify the robust way and, based on the statistical analysis performed, the inclusion of items on the scale (even those with a low factor), as well as the exclusion of others.

They present results of an original research and with high methodological rigor.

The discussion is well founded, supported by the data obtained through the results.

Conclusions lead us to the importance and relevance of using a valid and reliable scale as an instrument to be used in quality contraceptive counseling.

Most references are from the last 5 years.

The study respected the ethical principles required for studies with human beings: the authorizations of the University of California and the respective countries (Ethiopia and Egypt), as well as the informed consent given by the study participants.

I consider the article relevant for publication, due to the importance of dissemination, use of the scale and the methodological robustness presented.

• Thank you for this assessment of our paper; we appreciate your review and feedback.

---

## [Decision Letter · Decision Letter 1]

9 Nov 2022

PONE-D-21-36673R1Adaptation and validation of the quality of contraceptive counseling (QCC) scale for use in Ethiopia and IndiaPLOS ONE

Dear Dr. Holt,

Thank you for submitting your manuscript to PLOS ONE. After careful consideration, we feel that it has merit but does not fully meet PLOS ONE’s publication criteria as it currently stands. Therefore, we invite you to submit a revised version of the manuscript that addresses the points raised during the review process.

We look forward to receiving your revised manuscript.

Kind regards,

Wen-Jun Tu

Academic Editor

PLOS ONE

Journal Requirements:

Reviewers' comments:

Reviewer's Responses to Questions

**Comments to the Author**

1. If the authors have adequately addressed your comments raised in a previous round of review and you feel that this manuscript is now acceptable for publication, you may indicate that here to bypass the “Comments to the Author” section, enter your conflict of interest statement in the “Confidential to Editor” section, and submit your "Accept" recommendation.

Reviewer #1: All comments have been addressed

Reviewer #2: (No Response)

Reviewer #3: (No Response)

2. Is the manuscript technically sound, and do the data support the conclusions?

Reviewer #1: Yes

Reviewer #2: Yes

Reviewer #3: Yes

3. Has the statistical analysis been performed appropriately and rigorously? 

Reviewer #1: Yes

Reviewer #2: Yes

Reviewer #3: Yes

4. Have the authors made all data underlying the findings in their manuscript fully available?

Reviewer #1: Yes

Reviewer #2: Yes

Reviewer #3: Yes

5. Is the manuscript presented in an intelligible fashion and written in standard English?

Reviewer #1: Yes

Reviewer #2: Yes

Reviewer #3: Yes

6. Review Comments to the Author

Reviewer #1: The authors confused the use of exploratory factor analysis with confirmatory factor analysis. In one hand, they report that the aim of their work is tool validation which was supposed to be done via confirmatory analysis. Yet, the strategy utilized during their analysis was a typical characteristics of an exploratory analysis.

Reviewer #2: (No Response)

Reviewer #3: This is a thorough, well-done study. The reporting is clear. The inclusion of two different countries in the survey offers added value.

I have the following questions/comments:

1. Is there any information available about the response rate?

2. The factor analysis reveals that the QCC consists of 3 subscales. Does it make sense to calculate an overall QCC score (and an overall Cronbach's alpha)?

3. The item prov_insist is clearly measuring slightly differently from the other items in the "disrespect and abuse" subscale. I suggest to remove this item from the subscale. Since this item is considered important for the content validity, this item can still be included in the questionnaire, but it should not be used to calculate the score for “the disrespect and abuse” subscale.

7. PLOS authors have the option to publish the peer review history of their article (what does this mean?). If published, this will include your full peer review and any attached files.

Reviewer #1: No

Reviewer #2: **Yes: **Sara Elisabete Cavaco Palma

Reviewer #3: **Yes: **Wim Peersman

---

## [Author Response · Author response to Decision Letter 1]

3 Mar 2023

Reviewer #1: The authors confused the use of exploratory factor analysis with confirmatory factor analysis. In one hand, they report that the aim of their work is tool validation which was supposed to be done via confirmatory analysis. Yet, the strategy utilized during their analysis was a typical characteristics of an exploratory analysis.

Thank you for this comment. We have in fact conducted a confirmatory analysis, as detailed in the Analysis section beginning line 169.

Reviewer #2: (No Response)

Reviewer #3: This is a thorough, well-done study. The reporting is clear. The inclusion of two different countries in the survey offers added value.

I have the following questions/comments:

1. Is there any information available about the response rate?

No, we do not have response rates available. Sampling was done by convenience sampling, as detailed in the “Sample, recruitment, and data collection” section starting on line 117.

2. The factor analysis reveals that the QCC consists of 3 subscales. Does it make sense to calculate an overall QCC score (and an overall Cronbach's alpha)?

Thank you for this question. We argue that the Pearson correlation coefficients between subscale scores are high enough to suggest a single unifying latent QCC construct—as was similarly observed in the published Mexico version of the scale. The paragraph beginning line 308 provides the following detail: 

“Examination of the Pearson correlations coefficients between the subscale scores showed that the relationships observed in Mexico largely remained the same in Ethiopia, providing evidence for a single unifying latent QCC construct in this setting; specifically, the Information Exchange subscale and Interpersonal Relationship subscales were highly correlated (0.8 in Ethiopia, similar to the 0.7 observed in Mexico) and the Disrespect and Abuse subscale correlated 0.3 with both Information Exchange and Interpersonal Relationship in Ethiopia compared to the 0.3 and 0.4, respectively, observed in Mexico [11]. In India, the Information Exchange and Interpersonal Relationship subscales were correlated similarly to the other two countries (0.7); however, the Disrespect and Abuse subscale was less correlated with the other subscales (0.1 with Information Exchange and 0.05 with Interpersonal Relationship).” 

In the Discussion, we provide further guidance, noting the caveat that in India the D&A subscale was less correlated with the other subscales: 

“It is also worth noting that the Disrespect and Abuse subscale was less correlated with other subscales in our India sample compared to the Ethiopia sample and what had previously been observed in the QCC-Mexico validation study. This suggests that in the Vadodara context, the experience of Disrespect and Abuse is more weakly tied to the overall QCC construct; in other words, a person’s experience of extreme negative provider behaviors is less correlated with how they perceived counseling overall compared to other settings. Nonetheless, noting this limitation, composite scores can still be computed and applied to capture the full range of the QCC construct.”

 Further, the individual subscale alphas are also provided in Table 3.

3. The item prov_insist is clearly measuring slightly differently from the other items in the "disrespect and abuse" subscale. I suggest to remove this item from the subscale. Since this item is considered important for the content validity, this item can still be included in the questionnaire, but it should not be used to calculate the score for “the disrespect and abuse” subscale.

Thank you very much for this observation. It has motivated us to modify the analysis and move this item to the IR subscale where it performs much better. Because disrespect and abuse is highly related to the interpersonal relationship between provider and client, this change is justified conceptually. Line 276- footnote to Table 3-now reads: 

“This item [prov_insist] loaded as part of the Disrespect and Abuse subscale in the original scale development in Mexico [11]. In our CFA modeling it performed very poorly on that subscale (subscale alphas were 0.4 and 0.6 and the item loaded 0.02 and 0.14 in Ethiopia and India, respectively) and thus was moved to the conceptually-related Interpersonal Relationship subscale, as shown here”

Including the item in the IR subscale greatly improved the alphas for the D&A subscale in both countries and the loading of the item in India (though not Ethiopia, where it was nonetheless retained due to the critical importance of capturing potential pressure from providers from a content validity perspective). This change did not qualitatively change interpretation of findings related to goodness of fit or correlational validity, though the values were updated in the Results (Tables 3 & 4, respectively).

Other changes made:

• We caught a typo on line 100 about the length of the cognitive interviews in Ethiopia and updated to indicate that they were, on average, 30 minutes

• The QCC Website containing the item wording has been updated; accordingly, we replaced the old url with the new one in the text of the manuscript: https://qccscale.ucsf.edu/qcc-scale-versions

• In the Discussion, we referenced the short version of the QCC scale (the QCC-10); two new articles have been published so we added those citations to the reference section (19-20)

---

## [Decision Letter · Decision Letter 2]

22 Mar 2023

Adaptation and validation of the quality of contraceptive counseling (QCC) scale for use in Ethiopia and India

PONE-D-21-36673R2

Dear Dr. Holt,

We’re pleased to inform you that your manuscript has been judged scientifically suitable for publication and will be formally accepted for publication once it meets all outstanding technical requirements.

Kind regards,

Wen-Jun Tu

Academic Editor

PLOS ONE

Additional Editor Comments (optional):

Reviewers' comments:

Reviewer's Responses to Questions

**Comments to the Author**

1. If the authors have adequately addressed your comments raised in a previous round of review and you feel that this manuscript is now acceptable for publication, you may indicate that here to bypass the “Comments to the Author” section, enter your conflict of interest statement in the “Confidential to Editor” section, and submit your "Accept" recommendation.

Reviewer #3: All comments have been addressed

2. Is the manuscript technically sound, and do the data support the conclusions?

Reviewer #3: Yes

3. Has the statistical analysis been performed appropriately and rigorously? 

Reviewer #3: Yes

4. Have the authors made all data underlying the findings in their manuscript fully available?

Reviewer #3: Yes

5. Is the manuscript presented in an intelligible fashion and written in standard English?

Reviewer #3: Yes

6. Review Comments to the Author

Reviewer #3: The authors have adequately addressed my comments.

7. PLOS authors have the option to publish the peer review history of their article (what does this mean?). If published, this will include your full peer review and any attached files.

Reviewer #3: **Yes: **Wim Peersman

---

## [Editor Report · Acceptance letter]

24 Mar 2023

PONE-D-21-36673R2 

Adaptation and validation of the quality of contraceptive counseling (QCC) scale for use in Ethiopia and India 

Dear Dr. Holt:

I'm pleased to inform you that your manuscript has been deemed suitable for publication in PLOS ONE. Congratulations! Your manuscript is now with our production department. 

Kind regards, 

on behalf of

Dr. Wen-Jun Tu 

Academic Editor

PLOS ONE